# Anti-Thrombotic, Anti-Oxidant and Haemolysis Activities of Six Edible Insect Species

**DOI:** 10.3390/foods9040401

**Published:** 2020-04-01

**Authors:** Su-Jin Pyo, Deok-Gyeong Kang, Chuleui Jung, Ho-Yong Sohn

**Affiliations:** 1Department of Food and Nutrition, Andong National University, Andong 36729, Korea; vytn0608@naver.com (S.-J.P.); kangdk107@naver.com (D.-G.K.); 2Department of Plant Medicals, Andong National University, Andong 36729, Korea; cjung@andong.ac.kr

**Keywords:** edible insect, blood coagulation, platelet aggregation, haemolysis, *Teleogryllus emma*

## Abstract

In Korea, various insect species such as crickets and grasshoppers, as well as honey bee and silkworm pupae, have been consumed as food and used in oriental medicine. In this study to evaluate useful the bioactivities and potentially adverse effects of edible insects, ethanol extracts of *Allomyrina dichotoma* (AD), *Tenebrio molitor* (TM), *Protaetia brevitarsis* (PB), *Gryllus bimaculatus* (GB), *Teleogryllus*
*emma* (TE), and *Apis mellifera* (AM) were prepared and evaluated with regard to their anti-thrombosis, anti-oxidant and haemolysis activities against human red blood cells. AD and TE extracts showed strong anti-oxidant activities, which were not related to polyphenol content. All ethanol extracts, except AM extract, showed strong platelet aggregation activities. The platelet aggregation ratios of the extracts were 194%–246% of those of the solvent controls. The effects of the AD, TM, PB, GM, and AM extracts on thrombin, prothrombin and various coagulation factors were negligible. Only the extract of TM showed concentration-dependent anti-coagulation activities, with a 1.75-fold aPTT (activated Partial Thromboplastin Time) extension at 5 mg/mL. Of the six insect extracts, TM and AM extracts exhibited potent haemolytic activity. Our results on the insect extracts’ functional properties suggest that edible insects have considerable potential not just as a food source but as a novel bio-resource as well.

## 1. Introduction

As the imbalance between food production and consumption increases, the environmental burden to produce sufficient food creates diverse socio-economic concerns, compounded by climate change. To identify alternative food sources has thus become an important issue [1,2]. Insects are receiving particular attention as a sustainable alternative to meet future food demands and satisfy nutritional requirements [3,4,5,6,7]. It was estimated that at least 2000 species have been considered as edible by perhaps 2 billion people of various ethnic groups, representing approximately 30% of the global population [5]. 

In Korea, the use of insects in the traditional Korean medicine is an age-old practice [8,9] and certain species like silkworms (*Bombyx mori*) and grasshoppers (*Oxyoa sinuosa*) have been accepted as regular food items throughout the country [9,10,11,12]. Several studies have demonstrated the potential and commercial value of insects in terms of their protein content, composition of essential amino acids, fatty acids and mineral content [12,13]. The results came from analyses of three species of beetles (*Allomyrina dichotoma*: Dynastidae; *Protaetia brevitarsis*: Cetoniidae; *Tenebrio molitor*: Tenebrionidae), two species of crickets (*Teleogryllus emma* and *Gryllus bimaculatus*: Gryllidae) and the honey bee *Apis mellifera ligustica* [13,14]. Recently, the useful bioactivities of certain insects, such as the anti-adipogenic effect of *T. molitor* larvae in 3T3-L1 adipocytes and anti-obesity effects of *T. molitor* larvae in high-fat diet-induced obese mice [15], as well as the cytotoxic effect of *T. molitor* larvae extract against prostate (PC3 and 22Rv1), cervix (HeLa), liver (PLC/PRF5, HepG2, Hep3B, and SK-HEP-1), colon (HCT116), lung (NCI-H460), breast (MDA-MB231), and ovary (SKOV3) cancer cells [16], and lipid and homocysteine-lowering effects of *T. molitor* larvae in the plasma and liver of obese Zucker rats [17], were reported. In addition, the anti-microbial peptides from *A. mellifera* (royalisin), *Sacophaga peregrine* (sapecin), *Anopheles gambiae* and *B. mori* (defensin A, B, C) [18], the haemolysis activity of crude venom and mellitin from *A. mellifera* and wasp [19,20] and anti-exercise-fatigue activity of drone pupae extract of wasp in mice [21] were reported. However, limited information is available on their role as anti-oxidants and their bio-medical functional impact in connection with thrombosis-related issues and haemolysis remains unknown. 

In this study, the effects of the extracts of six edible insect species on blood coagulation, platelet-aggregation, and anti-oxidant activities were investigated to develop functional food ingredients using edible insects. Since adverse effects of insect extracts, such as anemia due to haemolysis, are well reported [19,20], haemolysis involving red blood cells of the insect extracts was also investigated. Our results regarding the insect extracts’ functional properties suggest that edible insects have considerable potential not just as a food source, but as a novel bio-resource as well. 

## 2. Materials and Methods 

### 2.1. Materials 

Samples of *Allomyrina dichotoma* (AD), *Tenebrio molitor* (TM), *Protaetia brevitarsis* (PB), *Gryllus bimaculatus* (GB), and *Teleogryllus emma* (TE) were obtained from commercial insect farms in Korea during May 2016, and sample of *Apis mellifera* (AM) was collected by A.B. Jessen, University of Copenhagen, Demark during the late summer in 2016 (Table 1). Details of the farming methods and culturing environments for the six edible insect species have previously been reported [13,14]. Insect samples were previously washed off by room-temperature water and dried. Not all insects were starved, except AD and PB, which were starved at the last instar for days to clean out the digestive tracks. The wings of honey bee drone adults were removed. All the samples were freeze-dried for at least 72 h at or below −50 °C, since killing insects with chemical or other physical means would have an influence on their bio-functionality. The dried samples were ground into powder and stored at −20 °C until further analyses. Reagent-grade chemicals were purchased from Fisher Scientific (Hampton, NH, USA). The blood samples were provided from Daegu-Gyeungbook Blood Center in Korea as Research blood. The study was conducted in accordance with the Declaration of Helsinki, and the protocol was approved by the Ethics Committee of Andong National University in Korea (IRB-Andong National Univ-482: 1040191-201602-BR-001-01, and IRB-Andong National Univ-1312: 1040191-202002-BR-001-01). Blood tests were conducted as a received state on the received day (same-day experiments), as recommended by suppliers. 

### 2.2. Sample Preparation 

To prepare ethanol extracts, 5 g of the lyophilized powders of the six edible insect species were dissolved with 100 mL of ethanol (95%, Daejung Chemicals & Metals Co., Ltd. Siheung, Korea) and extracted three times at room temperature. After that, the extract was filtered by paper (Whatsman No. 2) and concentrated under reduced pressure (Eyela Rotary evaporator, N-1000, Tokyo Rikakikai Co., Ltd., Tokyo, Japan). The powders of insects extracts were prepared using freeze dryer equipment (FD5508, Ilshin Lab Co. Ltd., Korea), and the insect samples were dissolved in dimethylsulfoxide (DMSO) at suitable concentrations for functional component analyses.

### 2.3. In-vitro Anti-Coagulation Activity 

The insect extracts were dissolved in DMSO, and anti-coagulation activity was measured by determining the clotting times required for the activation of thrombin (TT: Thrombin Time), prothrombin (PT: Prothrombin Time) and coagulation factors (aPTT: activated Partial Thromboplastin Time). TT, PT and aPTT were determined following the supplier’s instructions (Ameulung coagulometer, Lemgo, Germany) [22]. For the TT assay, 50 μL thrombin (0.5U, Sigma Co., St Louis, MO, USA), 50 μL CaCl_2_ (50 mM) and 10 μL insect extract were added to a preheated (37 °C) test tube and incubated at 37 °C for 3 min. Then, 100 μL preheated (37 °C) control plasma (MD Pacific Technology Co., Ltd., Huayuan Industrial Area, Tianjin, China) was added; the time (in seconds) from the addition of the control plasma to the detection of clot formation was measured and defined as the TT. For the PT assay, 70 μL control plasma and 10 μL insect extract were added to a preheated (37 °C) test tube and incubated at 37 °C for 3 min. Then, 130 μL preheated (37 °C) PT reagent (MD Pacific Technology Co., Ltd., Huayuan Industrial Area, Tianjin, China) was added; the time (in seconds) from the addition of PT reagent to the detection of clot formation was measured and defined as the PT. For the aPTT assay 70 μL control plasma and 10 μL insect extract were mixed in a preheated test tube at 37 °C for 3 min, and 65 μL preheated (37 °C) aPTT reagent (MD Pacific Technology Co., Ltd., Huayuan Industrial Area, Tianjin, China) was added and incubated at 37 °C for 3 min. Then, 65 μL CaCl_2_ (35 mM) preheated to 37 °C was added. The time (in seconds) from the addition of CaCl_2_ to the detection of clot formation was measured and defined as the aPTT [22]. Aspirin (1.5 mg/mL of acetylsalicylic acid, Sigma, St. Louis, MO, USA), which inhibits thrombin generation and clot formation at high doses (300–500 mg/day) [23], was used as positive control. All data are presented as the mean ± SD values of triplicates. 

### 2.4. In Vitro Platelet Aggregation Activity 

The effects of the insect extracts on platelet aggregation were measured using Whole Blood Aggregometer (Chrono-log, Havertown, PA, USA) [24]. The PRP obtained from human blood was washed with washing buffer, and the washed platelets (5 × 10^8^ cells/mL) were suspended using buffer containing 138 mM NaCl, 2.7 mM KCl, 12 mM NaHCO_3_, 0.36 mM NaH_2_PO_4_, 5.5 mM glucose, 0.49 mM MgCl_2_, and 0.25% gelatin (pH 7.4). The platelets were then incubated with the insect extracts, followed by stimulation with 2.5 μL of collagen (1 mg/mL) at 37 °C for 12 min. Impedance changes were monitored and amplitude, slope and area under the curve (AUC) were calculated using Aggrolink program (Aggrolink 5.2.3, Chrono-log, Havertown, PA, USA). Amplitude was expressed as ohms by the maximum extent of platelet aggregation and slope (rate of reaction) was determined by drawing a tangent through the steepest part of the curve. AUC was calculated from the platelet aggregation curve [22]. Platelet aggregation was calculated according to the following equation: (AUC of samples/AUC of DMSO) × 100 [24]. Aspirin, which inhibits the COX-1 and reduce thromboxane A2 synthesis at low concentration (100 mg/day) [25], was used as positive control of anti-platelet aggregation. 

### 2.5. Haemolysis Activity 

A human blood sample was diluted in phosphate-buffered saline (PBS; pH 7.4) and centrifuged at 200× *g* for 10 min. After three washes, the final concentration of the erythrocytes was adjusted to 4%. The erythrocyte suspension was transferred into 96-well plates and incubated with the insect extract at 37 °C for 1 h. The plate was centrifuged at 200× *g* for 10 min. The supernatant was collected and aliquoted and haemolysis was evaluated by determining the release of haemoglobin from the 4% human erythrocyte suspension at 414 nm with an ELISA reader. Zero percent and 100% haemolysis activities were evaluated with PBS alone and 0.1% Triton X-100, respectively. Haemolysis percentage was calculated using the following equation: haemolysis (%) = ((*A*_414 nm_ in the bee samples − *A*_414 nm_ in PBS)/(*A*_414 nm_ in 0.1% Triton X-100 − *A*_414 nm_ in PBS)) × 100 [26].

### 2.6. In-Vitro Anti-oxidant Activity 

DPPH (1,1-diphenyl-2-picryl hydrazyl) anion scavenging ability, ABTS (2,2-azobis(3-ethylbenzothiazoline-6-sulfonate)) cation scavenging activity, nitrite scavenging activity and reducing power were evaluated following previously reported methods [27,28]. Ascorbic acid (Sigma Co., St. Louis, MO, USA) was used as positive control. Each activity rating was expressed as the mean and the deviation of each of three replicates. 

DPPH radical scavenging activity of insect extracts was assessed according to the method described by Suzuki et al. [27]. The ability of insect extracts to scavenge DPPH free radicals was evaluated with reference to the results obtained for the ascorbic acid standard curve (0–1 mg 100 mL^−1^). The absorbance at 516 nm was measure using an Ultraviolet-visible spectrophotometer (Epouch Microplate reader, Biotek Instrument Inc., Winooski, VT, USA). 

ABTS•^+^ radical scavenging activity was measured using the ABTS radical cation decolorization assay, as described by Suzuki et al. [27]. The ability of insect extracts to scavenge ABTS•^+^ radicals was evaluated with reference to the results obtained for the ascorbic acid (Sigma Co., St. Louis, MO, USA) standard curve (0–1 mg 100 mL^−1^). The absorbance at 734 nm was measured using a UV-visible spectrophotometer (Epouch Microplate reader, Biotek Instrument Inc., Winooski, VT, USA).

The nitrite scavenging activity of the insect extracts was assessed according to the method described by Kim et al. [29] with some modification. A total of 5.0 mL insect extracts were mixed with sodium nitrite (1 mL, 0.01%) and citrate–phosphate buffer (pH 1.2) and adjusted to a volume of 10 mL with distilled water. The reaction solution was incubated at 37 °C for 1 h. Then, 3 mL of the reaction solution were mixed with 3 mL Griess reagent. After vigorous mixing with a vortex, the mixture was placed at room temperature for 15 min and the absorbance at 520 nm was recorded as A; B was the absorbance of the control group (water instead of sample solutions); C was the absorbance of the insect extract. Nitrite scavenging activity was calculated by the following equation: (1 − (A − C)/B) × 100 [27].

The reducing power of the insect extracts was assayed according to the method used by Jayaprakasha et al. [30] with some modifications. Briefly, 2.5 mL of insect extracts in ethanol were mixed with 2.5 mL of phosphate buffer (0.2 mol/L, pH 6.6) and 2.5 mL of potassium ferricyanide (10%). The mixtures were incubated for 20 min at 50 °C. Then, 2.5 mL of trichloroacetic acid (10%) and 0.5 mL FeCl_3_ (0.1%) were added to the mixture and incubated for 10 min; absorbance was measured at 700 nm against the buffer. Ascorbic acid was used as the standard.

### 2.7. Component Analysis 

The total polyphenol content of the six insect extracts was measured colorimetrically by using Folin-Ciocalteu’s phenol reagent [31]. Distilled water of 2.6 mL and 200 µL of Folin-Ciocalteu’s phenol reagent were added to 200 µL of the sample and mixed together; the mixture was allowed to react for 6 min at room temperature and then 2 mL of 7% (w/v) Na_2_CO_3_ solution was added. The mixture was then allowed to react for 30 min at 30 °C, and the absorbance was measured using a spectrophotometer (Epouch Microplate reader, Biotek Instrument Inc., Winooski, VT, USA) at 700 nm. A standard curve was constructed using rutin as the standard. 

The total flavonoid content in the six insect extracts was measured by the aluminium colorimetic method [32]. Distilled water of 320 µL and 15 µL of 5% (w/v) NaNO_2_ added to 100 µL of the sample were mixed together and allowed to react for 5 min. Then, 10% AlCl_3_ solution was added and the mixture was reacted for 1 min before 1M NaOH was added; subsequently, the absorbance was measured at 510 nm. A standard curve was constructed using tannic acid as a standard substance. Total sugar content was measured by the phenol–sulfuric acid method in microplate format [33], and the reducing sugar content was measured by the DNS method [34]. A standard curve was constructed using sucrose and glucose, respectively. 

### 2.8. Statistical Analysis 

Each activity value was expressed as the mean and the standard deviation of each of three replicates. Data were analyzed using SPSS packages (version 25). One-way analysis of variance (ANOVA) and Duncan Multiple Range Test at the level (*p <* 0.05) were used for the overall analysis of variance and mean separation, respectively.

## 3. Results

### 3.1. Component Anlysis of Ethanol Extracts 

The ethanol extracts for AD, TM. PB, GB, TE and AM were prepared and their extraction yields are shown in Table 2. The highest yield was observed in TM (40.5%) and followed by TE, PB, AD, GB and AM yields. The ethanol extraction yields were closely related to the content of crude lipid and oil of the insect samples [13,14]. Total polyphenol content showed that the GB and TE extracts possess 15.5–15.6 mg/g, whereas the AD extract has only 1.3 mg/g. Analysis of total flavonoid content showed that the AD extract possesses 5.7 mg/g, whereas the TM extract contains only 0.1 mg/g. Total sugar (89.3 ± 0.8 mg/g) and reducing sugar content (56.7 ± 1.5 mg/g) of the extract of AM were found to be the highest of the six insect extracts. 

### 3.2. In Vitro Anti-Coagulation Activity 

The anti-coagulation activities of the ethanol extracts of the six insect species were determined by measuring their TT, PT and aPTT. Treatment of aspirin (1.5 mg/mL) as positive control extended the clotting time to 1.64-fold of TT, 1.39-folds of PT, and 1.37-folds of aPTT compared with non-treatment (DMSO as solvent control) (Table 3). The treatment of high concentrations of aspirin (5 mg/mL) extended the clotting time to above 15-fold of TT, PT and aPTT compared with non-treatment. Treatment of the insect extracts (5.0 mg/mL) of AD, TM. PB, GB, and AM did not show significant changes in the clotting times of TT, PT and aPTT. However, the TE extract (5.0 mg/mL) was found to significantly extend TT (1.31-fold) and aPTT (1.75-fold) (Table 3), and the anti-coagulation effects of TE extract could be shown to be concentration-dependent (Figure 1).

### 3.3. In-vitro Platelet Aggregation Activities 

Since the platelet aggregation is strongly inhibited by aspirin [25], the changes in PAR (platelet aggregation ratio) by aspirin treatments were determined. At 0.25 and 0.125 mg/mL of aspirin, the PARs were 38.9% and 54.3% compared with non-treatment (Table 4, Figure 2). Strong and rapid activations of platelet aggregation were identified by treatment of the insect extract (0.25 mg/mL) of AD, TM, PB, GB and TE. The PARs of these insect extracts were 194.5%–246.1% compared with non- treatment. Only AM treatment (0.25 mg/mL) did not alter the platelet aggregation

### 3.4. Haemolytic Activities 

Haemolytic activities of the insect extracts were evaluated. Treatment with amphotericin B, known to possess potent haemolytic activity [26], and used as antifungal and anticancer agent, resulted in 53.4% haemolysis at 0.0125 mg/mL and 89.0% haemolysis at 0.025 mg/mL (Table 5). Among the insect extracts, AD, TM, PB and GB did not produce haemolysis up to 1.0 mg/mL (0.0–7.6% of haemolysis). However, the TE and AM extracts revealed potent haemolytic activity (94.5–97.5%) at 1.0 mg/mL. The calculated HC_50_s (the concentration for 50% haemolysis) for amphotericin B, TE and AM extracts were 0.015, 0.236 and 0.396 mg/mL, respectively. 

### 3.5. In-Vitro Anti-Oxdant Activities 

Oxidative stress is closely related to fibrin clot formation. Inappropriate posttranslational modification of fibrin and RBC haemolysis play roles in thrombotic disorders [35,36,37]. The assay of anti-oxidant activities showed that the AD and TE extracts (0.5 mg/mL) have potent DPPH anion scavenging power (59.9% for AD extract and 68.3% for TE extract), and ABTS cation scavenging activities (64.3% for AD extract and 62.1% for TE extract). The reducing powers of the AD, TE and AM extracts were found to be stronger than the extracts of TM, PB, GB and AM. Potent nitrite scavenging activity was observed in the PB and AM extracts.

## 4. Discussion

Insects and their solvent extractions have been used as medicinal resources and to promote health in Asian countries including China, Japan, and Korea and for a long time [38]. In Korea, the extract of PB has been used to treat inflammatory disease [39] and the extract of GB and *Aspongopus chinensis* has widely been used to treat inflammation [40,41]. Recent research demonstrated that various invertebrates have potent anti-thrombotic activities. The anti-thrombotic and fibrinolytic activities of *Lumbricus rubellus* [42] and anticoagulants from *Holotrichia diomphalia* larvae [43] are well reported. Furthermore, several anti-thrombotic compounds were identified, for example, the N-acetyltyramine, hydroxytyrosol and several diketopiperazines, which have anti-FXa and anti-platelet activities as in the TM [44], anti-platelet alkaloids from *Scolopendra subspinipes* [45], anti-platelet simplagrin from *Simulium nigrimanum* salivary glands [46], anticoagulant peptides from the mosquito *Culex pipiens* [47] and horsefly salivary glands [48], as well as *Bombyx batryticatus* [49]. However, assessments of the anti-thrombosis activities along with haemolytic and anti-oxidant activities of various edible insects are scarce. 

In this study, the high polyphenol contents (12.4–15.6 mg/g) and flavonoid contents (3.5–4.7 mg/g) were found in PB, GB, TE and AM extract (Table 2). However, the TM extract has 2.6 mg/g of polyphenol and 0.1 mg/g of flavonoid. The TM extract showed the lowest anti-oxidant activities among the insect extracts used. The contents of total sugar and reducing sugar in AM extract showed 3–10 folds higher than the other insect extracts. High level of sugars in AM would have been influenced by the feeding honey in the hive. However, the bee venom would not influence the results because we used only drones which lack the venom gland. Some insects also produce toxic substances such as benzoquinones from *Tenebrio molitor* under stress [50]. However, we tried to standardize the sampling and preparation process generic to all species we studied. 

We have determined the effects of the ethanol extracts of six edible insects (AD, TM, PB, GB, TE and AM) on blood coagulation and platelet aggregation. Extracts of these insect species have traditionally been considered as a part of anti-thrombotic oriental medicines in Korea. Only the TE extract showed significant and concentration-dependent inhibition against thrombin, prothrombin and blood coagulation factors (Table 3, Figure 1). However, the TE extract at 0.25 mg/mL showed the most potent platelet aggregation among the six insect extracts (Table 4, Figure 2). Considering the importance of platelet aggregation in thrombus formation as a primary haemostasis plug [24], the TE extract may result in accelerated thrombus formation. 

The TE and AM extracts had strong haemolytic activities against red blood cells (Table 5). Red blood cells (RBC) are the most abundant cell type in the blood and play an essential role in oxygen transfer. Several natural plant extracts and compounds of plants and/fungi, such as saponins, phenylhydrazine, penicillin, and cephalosporin induce haemolysis and result in erythrokatalysis [51,52,53,54]. Haemolysis of the RBC results in an exposure of intracellular phosphatidylserine, which accelerates blood coagulation and hyper-coagulable states [55]. Therefore, strong haemolysis activities of the TE extract may also contribute to the formation of fibrin clots. 

The extracts of AD and TE exhibit strong radical scavenging activities along with reducing powers (Table 6). Previous reports [35,36,37] have suggested that oxidative stress increases the modification of fibrin and RBC haemolysis. Thus, the strong anti-oxidant activities of these insect extracts may contribute to delayed blood clot formation, and ROS-induced cellular damage. However, our in vitro results on the coagulation and platelet aggregation assay did not suggest an anti-thrombotic effect of the insect extracts. Our results suggest that the ethanol extracts have haemostatic rather than anti-thrombotic activities. Further research on the purification of the thrombosis-related compounds and their underlying mechanism in vivo is required.

The potent nitrite scavenging activities of the PB and AM extracts are interesting (Table 6). Nitrite, as an important food additive in the meat processing industry, reacts with secondary amines to form nitrosoamine, which has emerged as a putative candidate responsible for colorectal cancer [56]. The consumption of excess amounts of nitrite could oxidize haemoglobin. The negative effects of nitrite on human health, including possible allergenic and vasodilator effects, were also reported [57]. Therefore, a reduction in nitrite exposure in protein-rich foods is helpful to reduce the development of colorectal cancer [56], and the idea of replacing nitrite with phytochemicals extracted from oregano or thyme provides an idea for the production of innovative processed meat [58]. However, there are no reports for the production of processed meat with PB and AM extracts to date Our results suggest the possible use of PB and AM extracts as anti-nitrite food ingredients in processed meat products, such as hams, sausages, bacons or crushed meats (Table 6). Furthermore, the extracts of PB did not show significant haeamolytic activity against RBC up to 1 mg/mL. Therefore, the evaluation of nitrite reduction and increased antioxidant of the PB extract-containing processed meat is worth further study. 

The antioxidant activities of the insect extracts are not related to the total polyphenol content (Table 2 and Table 6). The correlation coefficients of total polyphenol to DPPH scavenging, ABTS scavenging, nitrite scavenging, and reducing power were 0.003, 0.009, 0.125, and 0.094, respectively. The correlation coefficients of total polyphenol in respect to anti-coagulation activities (0.121 for TT, 0.387 for PT, and 0.085 for aPTT), and platelet aggregation rate (0.002) were not significant. These results suggest that phenolic compounds are not connected with anti-oxidant, anti-coagulation and platelet aggregation. It could be argued that edible insects like those examined in this paper could serve as a potential source of anti-oxidant ingredients. However, since the anti-oxidant activity was not related to the total polyphenol content, this suggests that edible insects are endowed with redox ingredients, ranging from phenolics and proteins to unidentified compounds, able to counteract oxidative stress from water and a lipophilic environment [59]. The purification of active compounds is in progress, and non-phenolic, lipophilic molecules may be related to the anti-thrombosis and anti-oxidant effects. 

## 5. Conclusions

Analyses of anti-coagulation, platelet aggregation, anti-oxidant and haemolytic activities of the six edible insects (*Allomyrina dichotoma*, *Tenebrio molitor*, *Protaetia brevitarsis*, *Gryllus bimaculatus*, *Teleogryllus emma* and *Apis mellifera*) have revealed that these insects have their own biological functionalities. Useful bioactivities of the six edible insects, such as the TE extract as an anti-coagulation, AD and TE extracts as anti-oxidation, PB and AM extracts as nitrite scavenging and TE and AM as hemolysis, were considered. The contents of polyphenol and flavonoids were not related to the antioxidant activity, which is dissimilar to the plant antioxidant system. All insect extracts except AM promoted platelet aggregation.

## Figures and Tables

**Figure 1 foods-09-00401-f001:**
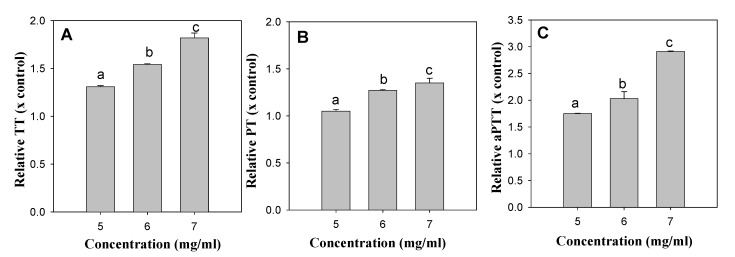
The concentration-dependent anti-coagulation activities of TE extract. (**A**): TT changes by TE extract, (**B**): PT changes by TE extract, (**C**): aPTT changes by TE extract. Different superscripts within a panel indicate statistically significant differences (*p* < 0.05).

**Figure 2 foods-09-00401-f002:**
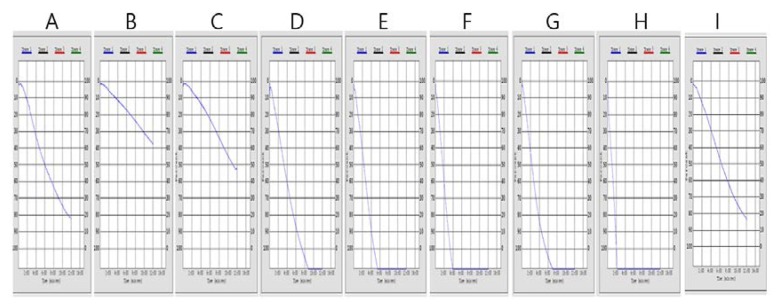
Impedance changes during platelet aggregation after addition of aspirin and the insect extracts in whole blood aggregometer. (**A**) DMSO, (**B**) aspirin (0.25 mg/mL), (**C**) aspirin (0.125 mg/mL), (**D**) AD extract (0.25 mg/mL), (**E**) TM extract (0.25 mg/mL), (**F**) PB (0.25 mg/mL), (**G**) GB extract (0.25 mg/mL), (**H**) TE extract (0.25 mg/mL) and (**I**) AM extract (0.25 mg/mL), respectively.

**Table 1 foods-09-00401-t001:** List of preferred insects used as food, feed, and oriental medicine in Korea.

Sample	Scientific Name	Classification	Feed	Life Stage
AD	*Allomyrina dichotoma*	Coleoptera Dynastidae	Fermented sawdust	Larvae
TM	*Tenebrio molitor*	Coleoptera Tenebrionidae	Rice, bran	Larvae
PB	*Protaetia brevitarsis*	Coleoptera Cetoniidae	Rice, bran, Organic soil	Larvae
GB	*Gryllus bimaculatus*	Orthoptera Gryllidae	Rice, bran, Vegetables	Late nymph
TE	*Teleogryllus emma*	Orthoptera Gryllidae	Rice, bran, Vegetables	Late nymph
AM	*Apis mellifera*	Hymenoptera Apidae	Honey, pollen	Adult

**Table 2 foods-09-00401-t002:** Yields and component assay of the ethanol extracts of 6 different insect species.

Extract	Yields (%)	Component (mg/g)			
		Total Polyphenol	Total Flavonoid	Total Sugar	Reducing Sugar
**AD**	23.1	1.3 ± 0.1 ^a^	5.7 ± 0.1 ^d^	13.0 ± 0.1 ^b^	1.8 ± 0.0 ^a^
TM	40.5	2.6 ± 0.2 ^b^	0.1 ± 0.1 ^a^	28.7 ± 1.0 ^c^	8.5 ± 0.2 ^c^
PB	24.9	11.8 ± 0.3 ^c^	3.5 ± 0.3 ^b^	31.5 ± 0.4 ^d^	13.0 ± 0.3 ^d^
GB	19.7	15.6 ± 0.3 ^d^	3.9 ± 0.3 ^b^	7.9 ± 0.3 ^a^	7.5 ± 0.3 ^c^
TE	30.1	15.5 ± 0.9 ^d^	4.7 ± 0.2 ^c^	8.5 ± 0.2 ^a^	6.1 ± 0.4 ^b^
AM	14.2	12.4 ± 0.2 ^c^	4.5 ± 0.3 ^c^	89.3 ± 0.8 ^e^	56.7 ± 1.5 ^e^

Different superscripts within a column indicate statistically significant differences (*p* < 0.05).

**Table 3 foods-09-00401-t003:** Effect of the ethanol extracts of 6 different insect species on blood coagulation.

Extract/Chemicals	Concentration (mg/mL)	Anti-Coagulation Activity (Multiplication of Control)		
		TT ^1^	PT ^2^	aPTT ^3^
**DMSO**	-	1.00 ± 0.07 ^bc^	1.00 ± 0.00^a^	1.00 ± 0.01 ^b^
Aspirin	1.5	1.64 ± 0.03 ^f^	1.39 ± 0.02^e^	1.37 ± 0.03 ^d^
	5.0	>15.0	>15.0	>15.0
AD	5.0	1.10 ± 0.01 ^d^	1.02 ± 0.02 ^ab^	1.11 ± 0.10 ^c^
TM	5.0	0.90 ± 0.00 ^a^	1.13 ± 0.03 ^c^	0.93 ± 0.01 ^ab^
PB	5.0	1.04 ± 0.03 ^c^	1.20 ± 0.01 ^d^	0.95 ± 0.01 ^ab^
GB	5.0	0.96 ± 0.00 ^b^	1.18 ± 0.00 ^d^	0.90 ± 0.02 ^a^
TE	5.0	1.31 ± 0.01 ^e^	1.05 ± 0.02 ^b^	1.75 ± 0.02 ^e^
AM	5.0	1.04 ± 0.04 ^c^	1.03 ± 0.04 ^ab^	0.96 ± 0.00 ^ab^

Anti-coagulation activity was calculated on the clotting time of a given sample divided by the clotting time of the solvent control in blood coagulation assays. The thrombin (TT^1^), prothrombin (PT^2^) and activated partial thromboplastin times (aPTT^3^) of the solvent control (DMSO) were 31.5 s, 20.5 s and 80.9 s, respectively. Data are means ± SD of triplicate determinations. Different superscripts within a column indicate statistically significant differences (*p* < 0.05).

**Table 4 foods-09-00401-t004:** Effect of the ethanol extracts of 6 different insect species on platelet aggregation.

Extract/Chemicals	Concentration (mg/mL)	Amplitude (ohm)	Slope	Lag Time (s)	Area Under Curve	PAR ^1^ (%)
DMSO	-	20	3	51	130.6	102.7
	-	20	3	59	123.7	97.3
Aspirin	0.25	9	1	60	49.4	38.9
	0.125	13	2	67	69.1	54.3
AD	0.25	27	4	17	215.3	194.5
TM	0.25	27	6	13	249.1	195.9
PB	0.25	26	8	8	262.5	206.4
GB	0.25	28	6	8	248.9	195.8
TE	0.25	25	12	1	272.4	246.1
AM	0.25	18	2	28	113.9	102.5

^1^ PAR: Platelet Aggregation Ratio. Data are presented as representative results relative to three independent determinations. Amplitude is expressed as ohms by maximum extent of platelet aggregation, and slope (rate of reaction) is determined by drawing a tangent through the steepest part of curve. Area under the curve (AUC) was calculated from the platelet aggregation curve.

**Table 5 foods-09-00401-t005:** Haemolytic activities of the ethanol extracts of 6 different insect extracts.

Extract/Chemicals	Concentration (mg/mL)	Haemolysis (%)
DMSO	-	0.0 ± 1.4 ^a^
Triton X-100	1.0	100.0 ± 0.4 ^e^
Amphotericin B	0.05	99.1 ± 0.4 ^e^
	0.025	89.0 ± 0.6 ^d^
	0.0125	53.4 ± 0.2 ^c^
	0.0063	1.1 ± 0.8 ^a^
	0.0032	0.0 ± 0.4 ^a^
AD	1.0	2.4 ± 0.5 ^a^
TM	1.0	7.6 ± 1.3 ^b^
PB	1.0	0.0 ± 3.5 ^a^
GB	1.0	1.1 ± 0.5 ^a^
TE	1.0	97.5 ± 2.2 ^e^
AM	1.0	94.5 ± 8.7 ^e^

Different superscripts within a column indicate statistically significant differences (*p* < 0.05).

**Table 6 foods-09-00401-t006:** Anti-oxidant activities of the ethanol extracts of 6 different insect species.

Extract	Scavenging Activity [SA] (%)			Reducing Power Abs. 700
	DPPH SA	ABTS SA	Nitrite SA	
AD	59.9 ± 0.8 ^e^	64.3 ± 2.5 ^d^	24.5 ± 0.7 ^ab^	0.269 ± 0.007 ^d^
TM	4.8 ± 1.7 ^a^	18.4 ± 1.5 ^a^	16.6 ± 0.2 ^a^	0.034 ± 0.008 ^a^
PB	15.9 ± 1.7 ^b^	32.8 ± 1.8 ^b^	46.8 ± 1.0 ^d^	0.151 ± 0.007 ^b^
GB	25.5 ± 0.6 ^d^	39.0 ± 0.9 ^c^	26.8 ± 1.4 ^b^	0.143 ± 0.003 ^b^
TE	68.3 ± 0.4 ^f^	62.1 ± 0.5 ^d^	20.6 ± 0.8 ^ab^	0.337 ± 0.009 ^e^
AM	18.5 ± 1.4 ^c^	40.1 ± 2.3 ^c^	40.4 ± 6.3 ^c^	0.230 ± 0.001 ^c^

The concentrations used for DPPH, ABTS, and reducing power assays were 500 μg/mL; for the nitrite scavenging activity assay 200 μg/mL were used. Different superscripts within a column indicate statistically significant differences (*p* < 0.05).

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
