# Peer review of "Anti-Thrombotic, Anti-Oxidant and Haemolysis Activities of Six Edible Insect Species"

_foods, 2020, doi:10.3390/foods9040401_

Round 1

Reviewer 1 Report

The manuscript evaluates the antithrombotic, antioxidative and haemolytic activities of six edible insect extracts. The topic of the paper is multi-faceted and interesting, however at present, the article contains few major flaws.

Ethics are the first and most important in my opinion. Despite the fact that freezing  is used in insect euthanasia, it is one of the least ethical options and should be used only if  chemical or instantaneous physical destruction is not possible (Recommendations for Ethical Euthanasia of Invertebrates (2.6 ed.). British and Irish Association of Zoos and Aquariums). I cannot firmly state that this is a mistake because there are no official guidelines for the welfare of insects but authors should take that into account. Moreover how many insects were used for research? What steps have been taken to use as few individuals as possible for research. On what basis was the number of insects estimated. Ethical issues related to human tissues (blood) are also important. Have the authors received the consent of the bioethics commission? Do the authors have the appropriate consent of the person from whom the blood was obtained for performing research and publication. The article lacks any detailed information about where this blood came from. There is also no precise description of who this sample was from or did it agree to use it for research? Was the sample only from one person? If so, then did the authors consider that blood from other people may react differently? Health status, allergies, diet or some diseases may affect the results of the tests. Could the blood group affect the results? What were the basic morphological parameters of tested blood?

What was the exact preparation of insects for experiments? Why the insects were not washed in some medium from breeding residues. Why was there no starvation diet (natural cleansing of the digestive tract)?. I am asking because, looking at table 2 in row AM, the statement that total sugar was about 89 mg / g, in my opinion rather indicates food residues in the digestive tract. Moreover more information is needed on what insects were fed. It should be noted that different diets can significantly affect the concentration of selected substances in the insect's body. I don't really understand why the bee wings were removed. For this I would understand, for example, the removal of the venom gland. Although bee venom itself is insoluble in alcohol, some of its components might. Also I would like to point out that some insects also produce toxic substances, e.g. Tenebrio molitor produces benzoquinones under stress. Benzoquinones have, for example, carcinogenic effects. Did the authors take this into account

In the article I also do not understand juggling the concentration of used extracts. Once a dose of 1mg / ml is tested, once a 0.25mg / ml, 5mg / ml. How were these doses determined? Materials and methods must clearly describe how these concentrations were calculated

There is also no exact description of the control sample. It should be clearly described in the materials and methods. Were blood tests performed without any additions? It should be included in every tested activities.

The discussion in my opinion contains a lot of speculation, not entirely supported by literature or experimental evidence. In the discussion there is a lack of broader description of  possible results use / exploitation in other areas

Finally, what I miss the most in this article is that these are in vitro tests, which should be clearly marked throughout the manuscript. I do not want to undermine the (more than)thousand-year old medicine,  but it focuses too much on the positive, leaving no room for potential negative effects. Entomotherapy is still considered as “alternative” medicine. I would not like such an interesting article to be used in pseudomedicine as a magic remedy, e.g. for cancer. I am asking you to refer more to the use of the potential activities in the materials / substances of the future medicine rather  than in therapy.

In my opinion, the article should be published, due to the innovative approach to the subject of edible insects, however, the authors must clarify the ambiguities and address some shortcomings

I also included some comments in the file

Kind regards

Author Response

Response to Reviewer 1 Comments

Firstly, we would like to thank the reviewers for their critical reviews on our manuscript and giving their valuable suggestions for its betterment.

We have addressed all the questions raised by the reviewers point by point.

Sincerely yours.

Reviewer 2 Report

Review

The authors applied a sound research approach in a relevant subject.

Minor revisions would include:

Line 172-173 Each activity rating value was expressed as the mean and the standard deviation of each of three replicates. Data were was analyzed

Line 191 …. of the ethanol extracts of for the 6 insect….

Line 196  …. and aPTT: and a PTT

Line 231-232  …. relative to independent three independent determinations

Line 236 Fig.2 Impedance…..Please correct: This should be located after the Figure NOT before the Fig as it is now.

Line 307   actibity   activity         1 mg/ml   1 mg/mL

Line 312  there is a comma missing after 0.009:  0.003, 0.009, 0.125 and …

Line 312  polyphenol to anti-coagulation  …. polyphenol in respect to anti-coagulation

Line 316  those examined by us in this paper could serve

Line 326 ……activities of the six edible insects (Allomyrina dichotoma, Tenebrio molitor, Protaetia brevitarsis, Gryllus bimaculatus, Teleogryllus emma and Apis mellifera) have revealed that these edible insects have

Line 329 Although TE and AM extracts showed haemolytic activities, the calculated HC50s of the six extracts suggested that the extracts of all six insect species extracts could be developed…..

Author Response

Response to Reviewer 2 Comments

Firstly, we would like to thank the reviewers for their critical reviews on our manuscript and giving their valuable suggestions for its betterment.

We have addressed all the questions raised by the reviewers point by point

The authors applied a sound research approach in a relevant subject.

Minor revisions would include:

1. Line 172-173 Each activity rating value was expressed as the mean and the standard deviation of each of three replicates. Data were was analyzed

Response 1: The mistake was corrected by reviewer’s comment.

2.Line 191 …. of the ethanol extracts of for the 6 insect….

Response 2: The mistake was corrected by reviewer’s comment.

3. Line 196  …. and aPTT: and a PTT

Response 3: The aPTT is the abbreviation of activated partial thromboplastin times.

4. Line 231-232  …. relative to independent three independent determinations

Response 4: The mistake was corrected by reviewer’s comment.

5. Line 236 Fig.2 Impedance…..Please correct: This should be located after the Figure NOT before the Fig as it is now.

Response 5: The mistake was corrected by reviewer’s comment.

6. Line 307   actibity   activity         1 mg/ml   1 mg/mL

Response 6: The mistake was corrected by reviewer’s comment.

7. Line 312  there is a comma missing after 0.009:  0.003, 0.009, 0.125 and …

Response 7: The mistake was corrected by reviewer’s comment.

8. Line 312  polyphenol to anti-coagulation  …. polyphenol in respect to anti-coagulation

Response 8: The mistake was corrected by reviewer’s comment.

9. Line 316  those examined by us in this paper could serve

Response 9: The mistake was corrected by reviewer’s comment.

10. Line 326 ……activities of the six edible insects (Allomyrina dichotoma, Tenebrio molitor, Protaetia brevitarsis, Gryllus bimaculatus, Teleogryllus emma and Apis mellifera) have revealed that these edible insects have

Response 10: The mistake was corrected by reviewer’s comment.

11. Line 329 Although TE and AM extracts showed haemolytic activities, the calculated HC50s of the six extracts suggested that the extracts of all six insect species extracts could be developed…..

Response 11: The mistake was corrected by reviewer’s comment.

Reviewer 3 Report

Foods-745476

This manuscript presents a study on the bioactivities and potentially adverse effects of edible insects. The authors do a good job of explaining the objectives, methods and results. Some minor corrections can be found below.

- Line 49- “haemolysis involvingt…”. The “t” should be removed.

- Line 72- “Freeze drying….” This line could be re-written for clarity.

- Line 91- “Then” is used a lot in this paragraph and is very repetitive for the reader.

Author Response

Response to Reviewer 3 Comments

Firstly, we would like to thank the reviewers for their critical reviews on our manuscript and giving their valuable suggestions for its betterment.

We have addressed all the questions raised by the reviewers point by point

This manuscript presents a study on the bioactivities and potentially adverse effects of edible insects. The authors do a good job of explaining the objectives, methods and results. Some minor corrections can be found below.

1. - Line 49- “haemolysis involvingt…”. The “t” should be removed.

Response 1: The mistake was corrected by reviewer’s comment.

2. - Line 72- “Freeze drying….” This line could be re-written for clarity.

Response 2: The mistake was corrected by reviewer’s comment.

: The powders of insects extracts were prepared using freeze dryer equipment (FD5508, Ilshin Lab Co. Ltd, Korea)

3. - Line 91- “Then” is used a lot in this paragraph and is very repetitive for the reader

Response 3: The mistake was corrected by reviewer’s comment.

For the aPTT assay 70 μL control plasma and 10 μL insect extract were mixed in a preheated test tube at 37 °C for 3 minutes, and 65 μL preheated (37 °C) aPTT reagent (MD Pacific Technology Co., Ltd, Huayuan Industrial Area, Tianjin, China) was added and incubated at 37 °C for 3 minutes.

Round 2

Reviewer 1 Report

the authors reliably referred to the reviews and corrected the flaws